# A Rare De Novo Mutation in the *TRIM8* Gene in a 17-Year-Old Boy with Steroid-Resistant Nephrotic Syndrome: Case Report

**DOI:** 10.3390/ijms25084486

**Published:** 2024-04-19

**Authors:** Marta Badeńska, Małgorzata Pac, Andrzej Badeński, Karolina Rutkowska, Justyna Czubilińska-Łada, Rafał Płoski, Nadezda Bohynikova, Maria Szczepańska

**Affiliations:** 1Department of Pediatrics, Faculty of Medical Sciences in Zabrze, Medical University of Silesia in Katowice, ul. 3 Maja 13/15, 41-800 Zabrze, Poland; marta.badenska2@gmail.com (M.B.); badenski.andrzej@gmail.com (A.B.); 2Department of Immunology, The Children’s Memorial Health Institute, 04-730 Warsaw, Poland; m.pac@ipczd.pl (M.P.); oddzial.immunologia@ipczd.pl (N.B.); 3Department of Medical Genetics, Medical University of Warsaw, 02-106 Warsaw, Poland; karolina.rutkowska@wum.edu.pl (K.R.); rploski@wp.pl (R.P.); 4Department of Neonatal Intensive Care and Neonatal Pathology, Faculty of Medical Sciences in Zabrze, Medical University of Silesia in Katowice, ul. 3 Maja 13/15, 41-800 Zabrze, Poland; jczubilinska.lada@gmail.com

**Keywords:** steroid-resistant nephrotic syndrome, children, dialysis, neurological complications, *TRIM8* gene mutation

## Abstract

Idiopathic nephrotic syndrome is the most common chronic glomerular disease in children. Treatment with steroids is usually successful; however, in a small percentage of patients, steroid resistance is observed. The most frequent histologic kidney feature of steroid-resistant nephrotic syndrome (SRNS) is focal segmental glomerulosclerosis (FSGS). Genetic testing has become a valuable diagnostic tool in defining the etiology of SRNS, leading to the identification of a genetic cause. The *TRIM8* gene is expressed in various tissues, including kidney cells and the central nervous system (CNS). An association between a mutation in the *TRIM8* gene and an early onset of FSGS has been proposed but is not well described. We present a 17-year-old boy with epilepsy, early mild developmental delay, a low IgG serum level, and proteinuria, secondary to FSGS. A Next-Generation Sequencing (NGS)-based analysis revealed a heterozygous de novo pathogenic variant in the *TRIM8* gene (c.1200C>G, p.Tyr400Ter). *TRIM8* gene sequencing should be considered in individuals with early onset of FSGS, particularly accompanied by symptoms of cortical dysfunction, such as epilepsy and intellectual disability.

## 1. Introduction

Idiopathic nephrotic syndrome (INS) is the most common chronic glomerular disease in children. It is characterized by proteinuria (24 h urine protein–creatinine ratio [PCR] ≥ 2 mg/mg), hypoalbuminemia, edema, and hyperlipidemia. It can be further complicated by hypertension, endocrine disorders, severe infections, and thrombotic events.

The prevalence of INS has reached 12–16 per 100,000 children aged under 16 [1]. In children and adolescents, nephrotic syndrome (NS) is classified depending on its response to a standardized steroid therapy into steroid-sensitive NS (SSNS) and steroid-resistant NS (SRNS). More than 85% of children with NS respond well to corticosteroids. Unfortunately, 10–15% remain unresponsive or later become steroid resistant. Such cases have been associated with an unfavorable renal prognosis. The most frequent histologic feature associated with SRNS is focal segmental glomerulosclerosis (FSGS), and, to a lesser extent, minimal change disease (MCD) or diffuse mesangial sclerosis (DMS) [2,3,4,5,6].

Genetic testing has become a valuable diagnostic tool in defining the etiology of SRNS, leading to the identification of a genetic cause in some cases. To date, over 30 monogenic mutations have been triggering SRNS [5,7], many of which implicate the glomerular podocyte and the slit membrane as the primary sites where the pathogenesis of SRNS unfolds [8]. The majority of genes known to cause SRNS are recessively inherited. Currently, there are no clear guidelines detailing the clinical utilization, relevance, and cost effectiveness of mutational screening for children with SRNS [9,10,11,12].

The *TRIM8* gene encodes the TRIM8 protein, a member of the tripartite motif (TRIM) protein family. The highest expression of the *TRIM8* gene has been noted in tissues of the central nervous system (CNS), kidney, and lens, with lower expression in the gut, as reported in in situ hybridization studies with mouse embryos [13]. De novo mutations of *TRIM8* have been reported in cases of childhood-onset epileptic encephalopathy and intellectual disabilities [14,15,16]. Few of those patients were reported to have concomitant proteinuria. One case manifested NS without neurological manifestations [17].

In this case report, the authors would like to present a rare de novo mutation in the *TRIM8* gene in a 17-year-old boy with SRNS.

## 2. Case Report

The boy was born to Caucasian parents after the second pregnancy and the second delivery, at the 38th week of gestational age. The delivery was uncomplicated, with a birth weight of 2450 g, a body length of 48 cm, and an APGAR score of 9.

### 2.1. Renal Manifestations

At the age of 10 years, the patient was admitted to the hospital due to proteinuria reaching 0.5 g/24 h. An abdomen ultrasound was performed, which revealed features of nephrocalcinosis with neither hypercalciuria nor signs of renal insufficiency. A month later, an angiotensin-converting enzyme (ACE) inhibitor was administered.

To establish the underlying pathology of the proteinuria, further tests were performed. The boy’s lipid profile was within the normal range, and the total serum protein estimation was at the lower limit (61 g/L). The level of complement component 3 (C3) was 0.94 g/L, which indicates a normal value, and complement component 4 (C4) reached 0.19 g/L. Antinuclear antibodies and antineutrophilic cytoplasmic autoantibodies were also negative. Therefore, vasculitis was no longer taken into consideration in the differential diagnosis.

At the age of 12, the patient underwent a renal biopsy, which confirmed FSGS. The hematoxylin and eosin (H&E)-stained slides showed 20 glomeruli in total, among which there were 2 glomeruli with well-defined peripheral segmental sclerosis, 10 glomeruli with global sclerosis, and 2 presenting signs of ischemia. No obliteration of capillary lumen was observed; however, there was an increase in the mesangial matrix, and 50% of the renal tubules were degenerated and atrophied.

Gradually, renal function started deteriorating, with a glomerular filtration rate (eGFR) at the level of 73.6 mL/min/1.73 m^2^ by the time the boy turned 13 (Figure 1). Initial genetic testing targeting hereditary glomerulopathies (PodoNet program: http://www.podonet.org/index.php?id=home, accessed on 18 April 2024) was performed, yet no mutation was confirmed (negative screening for the following: *NPHS1*, *NPHS2*, *PLCE1*, *LAMB2*, *SMARCAL1*, *ADCK4*, *COQ2*, *COQ6*, *PDSS2*, *MYO1E*, *PTPRO*, *GMS1*, *CD2AP*, *COL4A3/COL4A4*, *WT1*, *LMX1B*, *INF2*, *TRPC6*, *ACTN4*, and *COL4A5*). Urinary excretion of beta-2 microglobulin (B2M) remained within the normal range. No mutation in the *CLCN5* gene was found; therefore, Dent disease was also excluded.

Due to the constant decrease in renal functioning, it was decided to administer intravenous pulse corticosteroid therapy with Methylprednisolone (5 pulses per 500 mg) and later orally with Prednisone. Moreover, there was an attempt to include Cyclophosphamide (CP) in the therapy; however, it had to be withdrawn owing to the occurrence of side effects.

Despite the intensive steroid therapy, no remission was achieved. The last dose of steroids was administered when the patient was 15 years old. A little over a year afterwards, a Tenckhoff catheter was inserted, and the boy started peritoneal dialysis (PD) with no evidence of adverse side effects to date (Figure 2).

### 2.2. Neurological Manifestations

The patient experienced his first epileptic seizure at the age of 11. An electroencephalogram (EEG) showed paroxysmal disorders during spontaneous sleep. Control magnetic resonance imaging (MRI) of the head revealed no pathologies apart from a pineal gland cyst. Suitable therapy with valproic acid was prescribed. The patient had remained neurologically stable by the time the PD started. However, generalized tonic–clonic seizures appeared a week after the first dialysis, probably induced by hyponatremia. Head MRI revealed no pathologies; therefore, it was decided to increase the valproic acid dose up to 500 mg per day. The patient remains under constant neurological observation.

### 2.3. Psychological Evaluation

A psychological evaluation of the patient was conducted as well. The retardation of psychomotor development was established. The boy was described as socially withdrawn and deconcentrated, with separation anxiety and high emotional responsiveness. His intellectual or cognitive development was assessed as lower than average, with disharmonic course.

### 2.4. Immunological Evaluation

A laboratory evaluation of the patient’s immune system unveiled persistent hypogammaglobulinemia with low levels of serum immunoglobulin G (IgG) (2.76; 2.16; 3.08 g/L, N: 7.06–14.4 g/L) and serum immunoglobulin A (IgA) (0.15; 0.11; 0.13 g/L, N: 0.85–1.94 g/L). High urinary IgG excretion (114 mg/L, N: <14 mg/L) in the course of nephrotic proteinuria was also noted. The levels of IgM and the IgG3–4 subgroup were within the reference range or at its border; IgG1 and IgG2 subgroup levels were only moderately decreased.

Further examination revealed that C3 and C4 were within the normal range, and the presence of ANA antibodies was excluded.

Flow cytometric immunophenotyping of peripheral blood lymphocytes showed a normal percentage and normal absolute counts of CD4+, CD8+, CD19+, and natural killer (NK) cells with a normal CD4+/CD8+ ratio. The proportion of CD8+ naïve cells was slightly increased; the percentage of recent thymic immigrants was within the reference range. The defects in CD40 ligand expression were also excluded. An assessment of B lymphocyte maturation showed a normal percentage and normal absolute counts in the majority of the evaluated B-cell subsets, including memory B cells, with a decreased percentage of plasmablasts and memory B “switched” cells (Table 1).

To further assess cell-mediated immunity, we examined lymphocyte proliferation responses to the nonspecific mitogens (phytohemagglutynin (PHA), anty-CD3, and pansorbin). A lymphocyte transformation test (LTT) revealed normal indexes after stimulation.

In humoral response, the patient showed persistent protective levels of antibodies against diphtheria (0.82 IU/mL, N ≥ 0.1 IU/mL) and tetanus toxoid (1.27 IU/mL, N > 0.4 IU/mL). The anti-A isohemagglutinin titer was also normal.

Despite persistent hypogammaglobulinemia (deficiency of IgG), the patient had not suffered from frequent infections. Therefore, it was presumed that the immunologic abnormalities, observed in the course of nephrotic proteinuria, were the consequences of chronic end-stage renal disease. He was not qualified for immunoglobulin replacement therapy (IgRT).

We queried for possible genetic causes of syndromic features (encephalopathy and SRNS) revealing a novel *TRIM8* mutation. Other mutations seemed not to be connected to the patient’s problems.

### 2.5. Anamnesis Vitae

A pulmonary tract evaluation was conducted. Spirometry test results revealed a mild obstructive ventilatory defect (low forced expiratory volume/’forced/slow’ vital capacity (FEV1/FVC): 68.89; FEV1: 66% of the normal range; Maximal Expiratory Flow at 50% of FVC (MEF 50): 46% of the normal range). MRI and later computed tomography (CT) of the lungs exposed a few nodules placed in the lung fissures bilaterally, which were recognized as lymph nodes. No granulomatous changes were detected. No further examinations were considered vital; therefore, the boy remained under regular pulmonary observation.

The overall evaluation of the boy’s condition involved a gastroenterology consultation. It was driven to attention that he was rather slightly built and exhibited food reluctance. In view of the above, coeliac disease was suspected; however laboratory tests (Tissue Transglutaminase Antibodies (tTG-IgA and tTG-IgG)) were within the normal range.

An endocrinology consultation was performed as well. Medical history revealed that the patient was exhibiting signs of precocious puberty—isolated pubic hair development was observed. A pineal gland cyst (10 × 7 × 8 mm) was discovered during head MRI. Levels of testosterone, dehydroepiandrosterone sulfate (DHEAS), and androstenedione; a thyroid gland evaluation; and the diurnal cortisol profile were determined. There was a decrease in DHEAS; therefore, adrenal androgen deficiency was established. The patient underwent a check-up at the age of 16.5 years, which revealed normal gonadotropin and testosterone concentrations with slightly lower levels of DHEAS (still within the normal range).

In February 2022, the boy experienced acute pancreatitis (serum amylase: 1671 U/L, gamma-glutamyl transpeptidase (GGTP) 434 U/L), which was treated conservatively with a total recovery of pancreatic function.

## 3. Whole-Exome Sequencing

DNA samples from the proband, his parents, and his brother were isolated from peripheral blood using a Maxwell RSC Whole Blood DNA Kit (Promega, Madison, WI, USA) according to the manufacturer’s protocol. Isolated DNA from proband’s sample was used to prepare a whole-exome sequencing (WES) library with a Twist Human Core Exome spiked-in panel enriched in the following: the Twist mtDNA Panel, the Twist RefSeq Panel, and a Custom Panel covering variants located in noncoding regions that have been linked to clinical phenotypes according to the ClinVar database (Twist Bioscience, San Francisco, CA, USA). Enriched libraries were paired-end sequenced (2 × 100 bp) on NovaSeq 6000 (Illumina, San Diego, CA, USA). For the proband’s sample, 113,207,891 reads were obtained with a mean depth of 108.09 (99.5% of the target bases were covered at a minimum of 20×, whereas 99.7% of the bases had coverage of min. 10×). Raw WES data were bioinformatically analyzed, and variant prioritizations were performed as previously described [18]. After prioritization, a disease-causing variant segregation analysis was performed using amplicon deep sequencing (ADS) in the proband, his parents, and his brother with a Nextera XT Kit (Illumina, San Diego, CA, USA) and sequenced as was described for WES (Table 2).

In the proband’s sample, a nonsense de novo c.1200C>G variant in the *TRIM8* gene was identified (hg38, chr10:g.102656898-C>G, NM_030912.3, c.1200C>G, p.(Tyr400Ter)). ADS confirmed the presence of the heterozygous c.1200C>G variant in the proband, which was excluded in his parents and brother (Figure 3). The c.1200C>G variant causes a stop-gained change involving the alteration of a non-conserved nucleotide. In the ClinVar database (https://www.ncbi.nlm.nih.gov/clinvar, accessed on 18 April 2024), the c.1200C>G variant was reported as likely pathogenic. The in silico BayesDel addAF tool predicts a pathogenic outcome for this variant. The c.1200C>G variant was absent in the GnomAD database (v3.1.2; https://gnomad.broadinstitute.org/ accessed on 18 April 2024) and in the in-house database of >15,000 whole-exome sequences of Polish individuals.

## 4. Discussion

Symptoms occurring in the CNS, including seizures and delayed psychomotor development, have been well documented in children with *TRIM8* mutations. These mutations have been reported in autosomal dominant (AD) neuro-renal syndrome [19]. However, the function of TRIM8 proteins in podocytes is not well understood. Thus far, histopathological studies confirmed the diagnosis of FSGS in patients in the absence of proper Immunohistochemistry (IHC) staining of anti-TRIM8 antibodies. Therefore, it can be concluded that *TRIM8* is a novel gene associated with primary FSGS [16]. In the presented case, the c.1200C>G variant of the *TRIM8* gene was described in a patient with NS, seizures, delayed speech and language development, failure to thrive, short stature, and immunodeficiency disorder. The mutation was excluded in proband’s parents [20]. According to the OMIM (Online Mendelian Inheritance in Man; https://omim.org/about, accessed on 18 April 2024) database, expression of the pathogenic variants of the *TRIM8* gene (mainly nonsense and frameshift variants) lead to a focal segmental glomerulosclerosis and neurodevelopmental syndrome in an autosomal dominant inheritance pattern (MIM#619428).

TRIM8 proteins form homodimers. The molecular structure of TRIM8 comprises an N-terminal RING finger domain, type 1 and 2 B-box domains, a coiled coil domain, and a C-terminal region. The C-terminal region together with the RING motif are responsible for the interaction with SOCS1. The RING part transfers an E3 ubiquitin ligase activity for TRIM proteins. In a dose-dependent manner, TRIM8 lowers the level of the SOCS1 protein [21]. Further, the SOCS1 compound, by binding to phosphorylated tyrosine residues, prevents JAK-STAT signaling on the JAK proteins, which leads to the negative regulation of interferon signaling [22]. The coiled coil domain and the C-terminal region of TRIM8 are mandatory for homodimerization and nuclear localization [13].

The identified c.1200C>G variant was located in the last exon of the C-terminal proline-rich region of the *TRIM8* gene. According to the ClinVar database, pathogenic and likely pathogenic nonsense variants are exclusively found in the last exon of the *TRIM8* gene. It is likely that truncating mutations in the last exon of the *TRIM8* gene evade nonsense-mediated decay, leading to the generation of a shorter but stable product exhibiting a dominant-negative effect, especially since the TRIM8 functions as a homodimer [15].

The *TRIM8* expression is noted in the central nervous system, kidney, and eyes [23], implying that mutations may impair both renal and nervous system function. Several nonsense mutations in *TRIM8* have been described, causing various phenotypes (Figure 3C). Our c.1200C>G (p.Tyr400Ter) variant is situated in close proximity to the truncating c.1267C>T (p.Gln423Ter) variant reported in a 2-year-old patient presenting with global developmental delay, hypotonia, tonic–clonic seizures, and dysmorphic features. Another case reported nonsense variant c.1338T>A (p.Tyr446Ter) in a patient presenting with seizures, developmental delay, and the inability to sit, walk, or speak, along with NS [15]. Another study described an 8-year-old patient diagnosed with FSGS and neurodevelopmental syndrome with the c.1380T>A variant (p.Tyr460Ter). An analysis of kidney biopsy cells from the patient revealed no significant TRIM8 expression, indicating that the mutation likely disrupted the normal protein structure and homodimerization [16]. In a separate study, a 6-year-old boy exhibited prominent nephrotic-range proteinuria but with normal physical growth, intelligence, and motor development. This patient was identified as having the nonsense variant c.1484G>A (p.Trp495Ter). The c.1484G>A variant was positioned closest to the C-terminal region of the *TRIM8* gene [23], suggesting that renal lesions are more pronounced and neurological features are fewer in cases where mutations occur closer to the C-terminus. In summary, the phenotype resulting from truncating variants located at the end of the C-terminal part (last exon) of the *TRIM8* gene may differ from those occurring earlier in the exon. When mutations occur closest to the C-terminal region, the renal phenotype tends to be more severe compared to the neurological phenotype.

The c.1200C>G mutation may result in the production of both normal and non-functional TRIM8 dimers. The nonsense mutation may lead to the translation of a shorter protein that cannot be assembled into a homodimer, thereby decreasing the total amount of normal protein to about 25%, resulting in haploinsufficiency (the ratio of normal to altered protein being 1:3). Furthermore, a heterodimer created with a subunit containing the nonsense mutation may have an incorrect structure and be unable to function properly in the cell. Alternatively, in another scenario, the expressed heterodimer may exhibit a gain of function, resulting in a completely novel protein product or overexpression. Further studies are necessary to evaluate the effect of truncating *TRIM8* mutations on protein function.

The patient was diagnosed with precocious puberty with decreased DHEAS; therefore, adrenal androgen deficiency was established. The adrenocortical homeostasis was disrupted due to a heterozygous variant/mutation in the Armadillo-containing repeat protein 5 gene (*ARMC5*) [24]. In the literature, *ARMC5* mutations are proved to trigger various adrenocortical disorders resulting in a primary aldosteronism [25]. Moreover, it is believed that ARMC5 protein defects affect other cells, including the immune system, due to its influence on T-lymphocyte proliferation and differentiation [26].

The patient was diagnosed with an immune disorder owing to a decreased percentage of plasmablasts and memory B “switched” cells. The heterozygous variant/mutation in the mitogen-activated protein kinase kinase kinase kinase 5 (*MAP4K5*) gene was also detected. MAP4K5, a kinase homologous to SPS1/STE20 (KHS), expressed in various tissues and cells including T and B cells, is involved in the immune response regarding the CD40-mediated B-cell activity. The in vivo functions of KHS in the immune system still remain unclear. As KHS influences the c-Jun N-terminal kinase (JNK) activation in tumor necrosis factor α (TNFα) signaling, it might be involved in TNFα-associated specific inflammatory responses, such as those described in autoimmune inflammatory diseases, obesity, and type 2 diabetes [27].

Furthermore, the neurological impairment and heterozygous variant/mutation in the *GRID1* gene occurring in the presented case lead to the assumption of a significant role of glutamate receptor delta-1 subunit (GluD1), encoded by this gene, in the described pathologies. Gawande et al. obtained experimental results, which demonstrate GluD1 taking part in the regulation of the autophagy pathway. The presented outcome, proving a relevant genetic association between GluD1, hyperactivity, and cognitive deficits in mice, might confirm the influence of GluD1 on autism and other developmental disorders [28].

Niitsuma et al. were the first to identify two interleukin-1 receptor accessory protein (*IL1RAP*) variants/mutations in human siblings; therefore, they suggested that *IL1RAP* might be a causative gene for familial SSNS. When stimulated with IL-1β, peripheral blood mononuclear cells obtained from the siblings with SSNS produced significantly lower levels of cytokines as compared to healthy family members, showing an impairment of immune response [29]. The occurrence of a heterozygous variant/mutation in our patient sheds new light on the possible pathogenesis of SRNS in this case.

Cases reported in the literature suggest that patients with the *TRIM8* gene mutation who underwent renal transplantation are not likely to develop clinical or pathological FSGS recurrence in a long-term observation (up to 24 years) [30]. Therefore, kidney transplantation in our patient in the future may result in long-lasting improvement in the quality of life.

## 5. Conclusions

As the TRIM8 expression was found in the CNS, kidney, and eyes, it suggests that mutations may impair both renal and nervous system function. The nonsense mutation may lead to the translation of a shorter protein, resulting in haploinsufficiency. Moreover, heterodimers, created with such a subunit, may be unable to function properly in the cell.

Sequencing of the *TRIM8* gene should be considered in children with FSGS, especially in cases associated with epilepsy and intellectual disability. Since it is desired to provide a better understanding of pathologies in renal tissues and the CNS, further research regarding the pathogenesis of FSGS caused by a mutation in the *TRIM8* gene in children is still required.

## Figures and Tables

**Figure 1 ijms-25-04486-f001:**
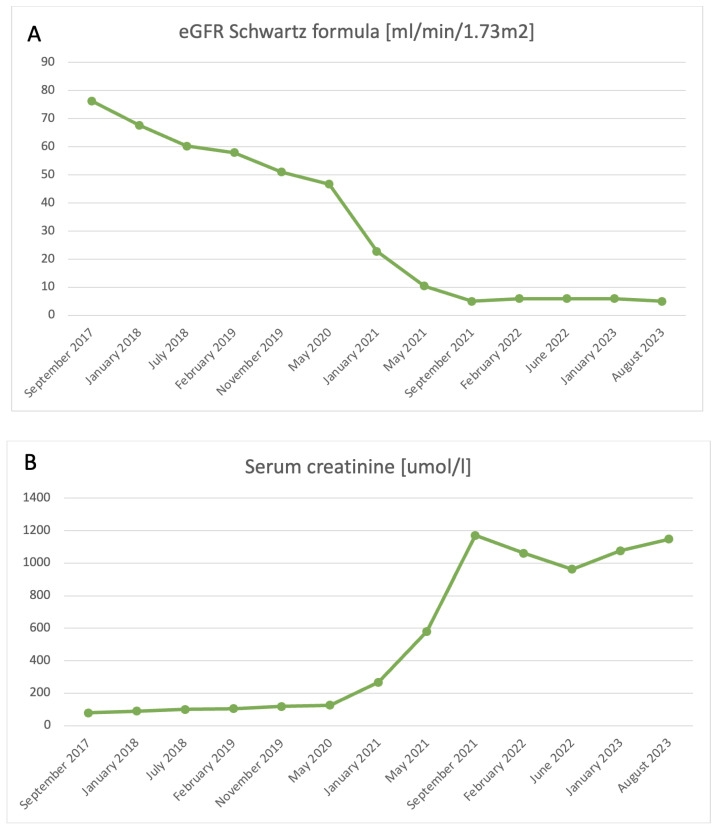
Fluctuations of selected laboratory parameters in time in the presented case. (**A**) Estimated glomerular filtration rate (eGRF) calculated with Schwartz formula. (**B**) Serum creatinine levels.

**Figure 2 ijms-25-04486-f002:**
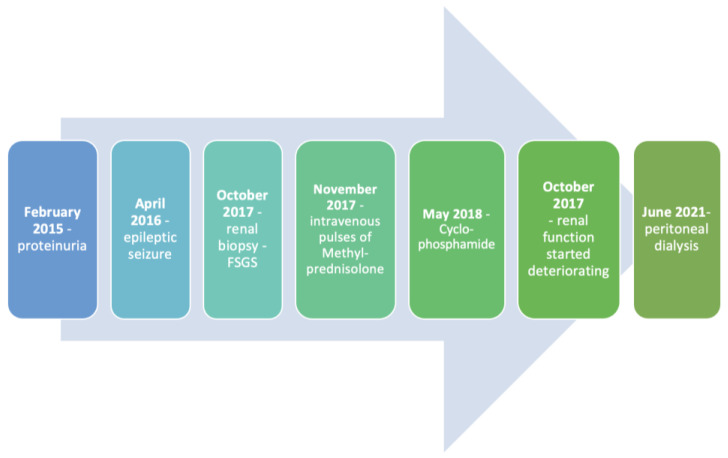
Timeline presenting crucial events in patient’s medical history.

**Figure 3 ijms-25-04486-f003:**
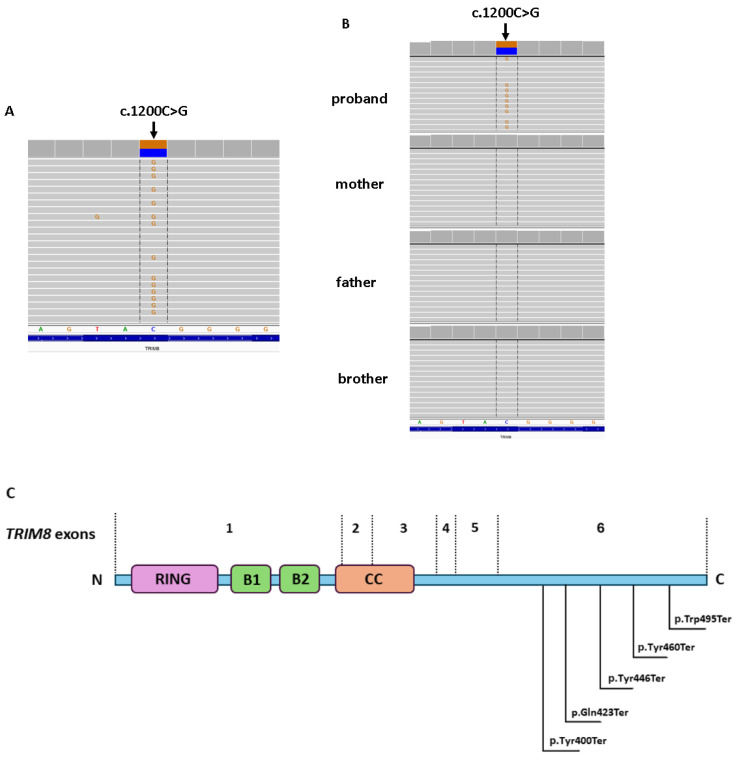
Genetic examination and family study of the c.1200C>G variant in the *TRIM8* gene presentation of the TRIM8 domains. (**A**) Shows WES results of proband. (**B**) Displays ADS of proband, his parents, and his brother. (Integrative Genomic Viewer is presented.) (**C**) Schematic presentation of the TRIM8 domains with the c.1200C>G variant and other variants previously reported.

**Table 1 ijms-25-04486-t001:** Flow cytometric immunophenotyping results.

Parameter		Percentage 1st Assessment	Percentage 2nd Assessment	Absolute Number (Cells/μL) 1st Assessment	Absolute Number (Cells/μL) 2nd Assessment	Normal Percentage Value	Normal Absolute Number (Cells/μL)
Lymphocytes		44	51.1			27.4–56.0	
Monocytes		7.8	7.3			6.1–10.5	
Granulocytes		48.3	41.6			36.2–64.9	
Lymphocytes	CD45+/SSC low			2748	3050		1500–3900
T Lymphocytes	CD3+/CD45+	81.5	78.1	2239	2381	52.9–79.1	1000–2700
Cytotoxic T cells	CD3+CD8+/CD45+	32.6	31.1	895	949	18.2–33.2	300–1100
Helper T cells	CD3+CD4+/CD45+	39.4	35.2	1083	1072	27.4–54.3	500–1600
Helper/cytotoxic T cells	CD3+CD4+CD8+/CD45+	1.4	0.4	37	13	0.5–1.8	10–50
NK cells	CD16+56+CD3-/CD45+	6.3	6.1	173	187	5.2–28.6	100–830
T cells with NK phenotype	CD3+CD16+CD56+/CD45+	4.9	4.3	134	132	1.2–6.9	27–164
B Lymphocytes	CD19+/CD45+	10.67	14.54	293	443	9.4–22.8	200–600
Helper/cytotoxic T-cell ratio	CD4+/CD8+	1.2	1.1			1.1–2.7	
Helper memory + effector memory lymphocytes	CD4+CD45RO+/CD3+CD4+	50.1	Not done	543	Not done	27.2–62.0	200–570
Helper naive + activated lymphocytes	CD4+CD45RA+/CD3+CD4+	50.5	Not done	547	Not done	31.1–66.3	180–880
Cytotoxic memory + effector memory lymphocytes	CD8+CD45RO+/CD3+CD8+	22.4	Not done	200	Not done	15.9–46.4	80–300
Cytotoxic naive + activated lymphocytes	CD8+CD45RA+/CD3+CD8+	81.1	Not done	726	Not done	44.1–77.1	170–730
Helper T cells							
Recent thymic emigrants	CD31+CD45RA+/CD3+CD4+	42.6	Not done	461	Not done	>30	
Central naive helper T cells	CD31-CD45RA+/CD3+CD4+	5.2	Not done	57	Not done		
Naive helper T cells	CD27+CD45RO-/CD3+CD4+	49.5	Not done	536	Not done	49.3–72.0	
Memory helper T cells	CD27+CD45RO+/CD3+CD4+	40.6	Not done	440	Not done	24.5–44.4	
Effector memory helper T cells	CD27-CD45RO+/CD3+CD4+	9.5	Not done	103	Not done	2.1–5.5	
Effector helper T cells	CD27-CD45RO-/CD3+CD4+	0.3	Not done	4	Not done	0.1–8.7	
Regulatory helper T cells	CD127-CD25+/CD3+CD4+	6.5	Not done	70	Not done	2.3–7.1	
Cytotoxic T cells							
Naive cytotoxic T cells	CD27+CD45RO-/CD3+CD8+	69	Not done	617	Not done	62.3–86.3	
Memory cytotoxic T cells	CD27+CD45RO+/CD3+CD8+	16.8	Not done	150	Not done	12.2–27.2	
Effector memory cytotoxic T cells	CD27-CD45RO+/CD3+CD8+	5.6	Not done	50	Not done	0.8–6.2	
Effector cytotoxic T cells	CD27-CD45RO-/CD3+CD8+	8.62	Not done	77	Not done	0.8–13.2	
Lymphocytes B							
Transitional B lymphocytes	CD38+IgM++/CD19+	10.49	6.31	31	28	1.4–13.0	10–60
Immature B lymphocytes	CD19+CD21low/limfocyty	1.11	0.74			0.6–1.9	
Immature B lymphocytes	CD21low/CD19+	10.4	5.09			2.9–13.2	10–50
Mature B lymphocytes	CD27-/CD19+	74	78.91	217	350	52.1–86.7	110–450
Naive mature B lymphocytes	IgD+cd27-/CD19+	65.19	76.15	191	338	51.3–82.5	120–430
Memory B lymphocytes	CD27+/CD19+	26	21.09	76	94	13.3–47.9	50–200
Memory “non-switched” B lymphocytes (marginal zone)	CD27+IgD+/CD19+	18.04	16.04	53	71	4.6–18.2	20–70
Memory “switched” B lymphocytes	CD27+IgD-/CD19+	7.85	5.09	23	23	8.7–25.6	30–110
Memory “switched” B lymphocytes	CD27+IgD-/PBL	0.84	0.74			1.5–4.0	
IgM-only memory B lymphocytes	IgD-IgM+/CD19+IgD-CD27+	39.27	15.31			3.9–18.8	
IgM-only memory B lymphocytes	IgD-IgM+CD27+/CD19+	3.08	0.78			0.4–3.1	
Activated B lymphocytes	CD38lowCD21low/CD19+	7.94	2.1	23	9	2.7–8.7	10–40
Plasmablasts	CD38+++IgM-/CD19+	0.4	0.21	1	1	0.6–6.5	0–20
CD40 Ligand							
Stimulated	CD154+/CD3+CD8-	51.5					
Stimulated	CD69+/CD3+CD4+	99.5					

**Table 2 ijms-25-04486-t002:** Presence of gene variants in individual family members.

Gene	Variant	Proband	Mother	Father	Brother
*GRID1*	c.2771del	Heterozygous	Excluded	Heterozygous	Heterozygous
*MAP4K5*	c.2293G>C	Heterozygous	Heterozygous	Excluded	Heterozygous
*TRIM8*	c.1200C>G	Heterozygous	Excluded	Excluded	Excluded
*IL1RAP*	c.911A>G	Heterozygous	Heterozygous	Excluded	Excluded
*ARMC5*	c.49G>C	Heterozygous	Heterozygous	Excluded	Heterozygous

## Data Availability

Data are contained within the article.

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
