# Peer review of "A Rare De Novo Mutation in the TRIM8 Gene in a 17-Year-Old Boy with Steroid-Resistant Nephrotic Syndrome: Case Report"

_ijms, 2024, doi:10.3390/ijms25084486_

Round 1
Reviewer 1 Report
Comments and Suggestions for Authors
The case has a limited impact because patients with similar mutations (inducing truncations in the same region of the gene) have been already described by others. No attempt is made at understanding, or at least describing, how the truncation is affecting the TRIM8 structure and function and how this would induce the reported pathological effects. I found several points that must be addressed prior to re-consider this report for publication.
_In Table 1 a long list of immunological parameters have been reported while a series of neurological, psychological, pulmonary, endocrine and gastroenterological symptoms are only described with no attempt at showing anything in a more quantitative fashion. This section should be improved reporting some of the collected clinical data in another table or figure.
_As stated by the Authors (refs 14-17), a series of mutations (causing truncations) affecting the C-terminal domain of TRIM8 have been already described. Therefore, the impact and novelty of the new case that they have found remains limited. Ideally, these other sites should be reported in the scheme suggested in the point below. See for example Fig.3 in “De novo truncating variants of TRIM8 and atypical neuro-renal syndrome: a case report and literature review” by Li and Guo, Italian Journal of Pediatrics (2023) 49:46 https://doi.org/10.1186/s13052-023-01453-4.
_A scheme of the domain architecture of human TRIM8 should be added to Fig.1, in which also Sanger sequencing results should be reported showing the position of the nonsense codon and how it would affect the last C-terminal domain. In fact, it remains highly unclear how exactly the truncation would affect the structure and function of the enzyme
_A relevant question related to the previous point: is the position of the truncation within the C-terminal domain of TRIM8 related to the severity of the phenotype observed? Could it be that if the truncation site is in a location more towards the C-terminal end of the protein then the phenotype observed would be milder? I believe that this is an interesting point that should be discussed.
_Was it possible to analyze TRIM8 by Western blots of total protein extracts obtained from a biopsy?
_line 226: “TRIM8 proteins play the role of homodimers”. Please re-phrase.
_Is this a case in which a “dominant negative” effect is observed or, being TRIM8 a homodimer, the observed pathological effects would depend on TRIM8 haploinsufficiency? In a heterozygous patients, can one assume that a certain amount of unaffected homodimers, although maybe reduced, would be formed?
_All in all, based on the data presented it remains highly unclear how the observed truncation of TRIM8 would cause the pathological effects recorded.
Comments on the Quality of English LanguageI have no comments
Author Response
The case has a limited impact because patients with similar mutations (inducing truncations in the same region of the gene) have been already described by others. No attempt is made at understanding, or at least describing, how the truncation is affecting the TRIM8 structure and function and how this would induce the reported pathological effects. I found several points that must be addressed prior to re-consider this report for publication.
_In Table 1 a long list of immunological parameters has been reported while a series of neurological, psychological, pulmonary, endocrine and gastroenterological symptoms are only described with no attempt at showing anything in a more quantitative fashion. This section should be improved reporting some of the collected clinical data in another table or figure.
Answer: Thank you for the comment. The authors added a timeline of collected clinical data (Figure 2).
_As stated by the Authors (refs 14-17), a series of mutations (causing truncations) affecting the C-terminal domain of TRIM8 have been already described. Therefore, the impact and novelty of the new case that they have found remains limited. Ideally, these other sites should be reported in the scheme suggested in the point below. See for example Fig.3 in “De novo truncating variants of TRIM8 and atypical neuro-renal syndrome: a case report and literature review” by Li and Guo, Italian Journal of Pediatrics (2023) 49:46 https://doi.org/10.1186/s13052-023-01453-4.
Answer: Thank you for your suggestion. Variants previously reported in the literature were added to the scheme (Figure 3C).
_A scheme of the domain architecture of human TRIM8 should be added to Fig.1, in which also Sanger sequencing results should be reported showing the position of the nonsense codon and how it would affect the last C-terminal domain. In fact, it remains highly unclear how exactly the truncation would affect the structure and function of the enzyme
Answer: Thank you for your suggestion. A schematic presentation of TRIM8 domains was added (Figure 3C). Because of the short time assigned for review we are not able to perform Sanger sequencing. However, please note that variant has been verified by amplicon NGS sequencing, which currently is the standard in our lab.
The question how the position of the nonsense codon would affect the last C-terminal domain was explained in the discussion as follows:
“The identified c.1200C>G variant was located in the last exon of the C-terminal proline-rich region of the TRIM8 gene. According to the ClinVar database, pathogenic and likely pathogenic nonsense variants are exclusively found in the last exon of the TRIM8 gene. It is likely that truncating mutations in the last exon of the TRIM8 gene evade nonsense-mediated decay, leading to the generation of shorter but stable product exhibiting dominant-negative effect, especially since the TRIM8 functions as homodimer (Assoum et al. 2017).”
_A relevant question related to the previous point: is the position of the truncation within the C-terminal domain of TRIM8 related to the severity of the phenotype observed? Could it be that if the truncation site is in a location more towards the C-terminal end of the protein then the phenotype observed would be milder? I believe that this is an interesting point that should be discussed.
Answer: Thank you for your question. The TRIM8 expression is noted in the central nervous system, kidney, and eyes (Wei et al., 2023), implying that mutations may impair both renal and nervous system function. Several nonsense mutations in the TRIM8 have been described causing various phenotypes (Fig. 1C). Our c.1200C>G (p.Tyr400Ter) variant is situated in close proximity to the truncating c.1267C>T (p.Gln423Ter) variant reported in a 2-year-old patient presenting with global developmental delay, hypotonia, tonic–clonic seizures, and dysmorphic features. Another case reported a nonsense variant c.1338T>A (p.Tyr446Ter) in a patient presenting with seizures, developmental delay, inability to sit, walk, or speak, along with nephrotic syndrome (Assoum et al. 2017). Another study described a 8-year-old patient diagnosed with focal segmental glomerulosclerosis and neurodevelopmental syndrome with c.1380T>A variant (p.Tyr460Ter). Analysis of kidney biopsy cells from the patient revealed no significant TRIM8 expression, indicating that the mutation likely disrupted the normal protein structure and homodimerization (Warren et al., 2020). In a separate study, a 6-year-old boy exhibited prominent nephrotic-range proteinuria but with normal physical growth, intelligence, and motor development. This patient was identified with the nonsense variant c.1484G>A (p.Trp495Ter). The c.1484G>A variant was positioned closest to the C-terminal region of the TRIM8 gene (Wei et. al., 2023), suggesting that renal lesions are more pronounced and neurological features are fewer in cases where mutations occur closer to the C-terminus. In summary, the phenotype resulting from truncating variants located at the end of the C-terminal part (last exon) of the TRIM8 gene may differ from those occurring earlier in the exon. When mutations occur closest to the C-terminal region, the renal phenotype tends to be more severe compared to the neurological phenotype.
_Was it possible to analyze TRIM8 by Western blots of total protein extracts obtained from a biopsy?
Answer: Thank you for the question. Unfortunately, such analysis wasn’t possible.
_line 226: “TRIM8 proteins play the role of homodimers”. Please re-phrase.
Answer: Thank you for the suggestion. The sentence was changed into “TRIM8 proteins form homodimers.”
_Is this a case in which a “dominant negative” effect is observed or, being TRIM8 a homodimer, the observed pathological effects would depend on TRIM8 haploinsufficiency? In a heterozygous patients, can one assume that a certain amount of unaffected homodimers, although maybe reduced, would be formed?
Answer: Thank you for your question. The c.1200C>G mutation may result in the production of both normal and non-functional TRIM8 dimers. The nonsense mutation may lead to the translation of a shorter protein that cannot be assembled into a homodimer, thereby decreasing the total amount of normal protein to about 25%, resulting in haploinsufficiency (the ratio of normal to altered protein being 1:3). Furthermore, heterodimer created with a subunit containing the nonsense mutation may have an incorrect structure and be unable to function properly in the cell. Alternatively, in another scenario, the expressed heterodimer may exhibit a gain of function, resulting in a completely novel protein product or overexpression. Further studies are necessary to evaluate effect of truncating TRIM8 mutations on protein function.
_All in all, based on the data presented it remains highly unclear how the observed truncation of TRIM8 would cause the pathological effects recorded.
Answer: Whereas we agree that more experiments are needed, the answer to this question is to some extent provided in the Discussion above.

Reviewer 2 Report
Comments and Suggestions for Authors
This is an interesting article by Marta Badeńska et al.
I would like to address a small number of suggestions to you.
General recommendations
Please correct all references according to journal stile.
References should be described as follows:
Journal Articles:
1. Author 1; Author 2. Title of the article. Abbreviated Journal Name Year; Volume: page range.
Page 2, line 53
The term "central nervous tissue" is wrong.
Page 2, line 55 and line 59
Please write "De novo" in italics.
Page 2, line 58
Please correct "with no" to "without"
Page 2,
The title " Clinical report" may write as "Case report".
Authors start this case report with data about the gestational history.
Is this information plays critical role for the disease formation?
Page 2, line 66 Please change the title "Renal problems" to “Renal manifestation”.
Page 3, line 100
Please change the title "Neurological problems" to “Neurological manifestation".
Page 3, line 101
It is not fully correct to start the section with phrase that patient remained under constant neurological observation, without dentally description of neurological manifestation. This sentence may finalize this paragraph.
Page 3, section "2.4. Immunological evaluation"
Considering analytical description of immune status in this section, I suggest writing all immunological examination including the complement and ANA antibodies in this section.
It is not fully understanding why appendicitis with appendix removal and COVID-9 have been referred by authors as separate section.
I suggest adding all patients' history except from renal and neurological involvement as separate section "Anamnesis vitae" (Past medical history).
Page 9 line 247
Please write in vivo in italics.
Conclusions are too short.
Author Response
This is an interesting article by Marta Badeńska et al.
I would like to address a small number of suggestions to you.
General recommendations
Please correct all references according to journal stile.
References should be described as follows:
Journal Articles:
- Author 1; Author 2. Title of the article. Abbreviated Journal NameYear; Volume: page range.
Answer: The references were corrected.
Page 2, line 53
The term "central nervous tissue" is wrong.
Answer: Thank you for your suggestion. The term was changed into “tissues of central nervous system”.
Page 2, line 55 and line 59
Please write "De novo" in italics.
Answer: The corrections were made.
Page 2, line 58
Please correct "with no" to "without"
Answer: The corrections were made.
Page 2,
The title " Clinical report" may write as "Case report".
Answer: The corrections were made.
Authors start this case report with data about the gestational history.
Is this information plays critical role for the disease formation?
Answer: Thank you for your question. Gestational history was mentioned in the manuscript to exclude the possible influence of fetal distress or birth defect on the described condition.
Page 2, line 66 Please change the title "Renal problems" to “Renal manifestation”.
Answer: The corrections were made.
Page 3, line 100
Please change the title "Neurological problems" to “Neurological manifestation".
Answer: The corrections were made.
Page 3, line 101
It is not fully correct to start the section with phrase that patient remained under constant neurological observation, without dentally description of neurological manifestation. This sentence may finalize this paragraph.
Answer: Thank you for the suggestion. The corrections were made.
Page 3, section "2.4. Immunological evaluation"
Considering analytical description of immune status in this section, I suggest writing all immunological examination including the complement and ANA antibodies in this section.
Answer: Thank you for the comment. The information was added in the manuscript.
It is not fully understanding why appendicitis with appendix removal and COVID-9 have been referred by authors as separate section.
Answer: Thank you for the suggestion. These sections were removed.
I suggest adding all patients' history except from renal and neurological involvement as separate section "Anamnesis vitae" (Past medical history).
Answer: Thank you for the suggestion. The corrections were made.
Page 9 line 247
Please write in vivo in italics.
Answer: The corrections were made.
Conclusions are too short.
Answer: The conclusions has been rewritten.

Round 2
Reviewer 1 Report
Comments and Suggestions for Authors
The Authors have properly addressed some of the points I raised. I believe the manuscript is improved.